# SQLNet: Generating Structured Queries From Natural Language Without Reinforcement Learning

## Abstract

Synthesizing SQL queries from natural language is a long-standing open problem and has been attracting considerable interest recently. Toward solving the problem, the de facto approach is to employ a sequence-to-sequence-style model. Such an approach will necessarily require the SQL queries to be serialized. Since the same SQL query may have multiple equivalent serializations, training a sequence-to-sequence-style model is sensitive to the choice from one of them. This phenomenon is documented as the "order-matters" problem. Existing state-of-the-art approaches rely on reinforcement learning to reward the decoder when it generates any of the equivalent serializations. However, we observe that the improvement from reinforcement learning is limited.

In this paper, we propose a novel approach, i.e., **SQLNet**, to fundamentally solve this problem by avoiding the sequence-to-sequence structure when the order does not matter. In particular, we employ a sketch-based approach where the sketch contains a dependency graph so that one prediction can be done by taking into consideration only the previous predictions that it depends on. In addition, we propose a sequence-to-set model as well as the column attention mechanism to synthesize the query based on the sketch. By combining all these novel techniques, we show that **SQLNet** can outperform the prior art by $9\%$ to $13\%$ on the WikiSQL task.

## 1 Introduction

Semantic parsing is a long-standing open question and has many applications. In particular, parsing natural language descriptions into SQL queries recently attracts much interest from both academia (Yaghmazadeh et al., 2017) and industry (Zhong et al., 2017). We refer to this problem as the natural-language-to-SQL problem (NL2SQL). The de facto standard approach to solve this problem is to treat both the natural language description and SQL query as sequences and train a sequence-to-sequence model (Vinyals et al., 2015b) or its variants (Dong & Lapata, 2016) which can be used as the parser. One issue of such an approach is that different SQL queries may be equivalent to each other due to commutativity and associativity. For example, consider the following two queries:

```
SELECT result                SELECT result
WHERE score='1-0' AND goal=16   WHERE goal=16 AND score='1-0'
```

The order of the two constraints in the `WHERE` clause does not affect the execution results of the query, but syntactically, these two are considered as different queries. It is well-known that the order of these constraints affects the performance of a sequence-to-sequence-style model (Vinyals et al., 2016), and it is typically hard to find the best ordering. To mitigate this ordering issue, a typical approach that has been applied in many scenarios is to employ reinforcement learning (Zhong et al., 2017; Hu et al., 2017). The basic idea is that, after a standard supervised training procedure, the model is further trained using a policy gradient algorithm. In particular, given an input sequence, the decoder of a sequence-to-sequence model samples an output sequence following the output distribution and computes the reward based on whether the output is a well-formed query and whether the query will compute the correct results. This reward can be used by the policy gradient algorithm

to fine-tune the model. However, the improvement that can be achieved through reinforcement learning is often limited. For example, on a NL2SQL task called WikiSQL (Zhong et al., 2017), the state-of-the-art work (Zhong et al., 2017) reports an improvement of only $2\%$ by employing reinforcement learning.

In this work, we propose SQLNet to fundamentally solve this issue by avoiding the sequence-to-sequence structure when the order does not matter. In particular, we employ a sketch-based approach to generate a SQL query from a *sketch*. The sketch aligns naturally to the syntactical structure of a SQL query. A neural network, called SQLNet, is then used to predict the content for each *slot* in the sketch. Our approach can be viewed as a neural network alternative to the traditional sketch-based program synthesis approaches (Alur et al., 2013; Solar-Lezama et al., 2006; Bornholt et al., 2016; Rabinovich et al., 2017; Parisotto et al., 2017). Note that the-state-of-the-art neural network SQL synthesis approach (Zhong et al., 2017) also employs a sketch-based approach, although their sketch is more coarse-grained and they employ a sequence-to-sequence structure to fill in the most challenging slot in the sketch.

As discussed above, the most challenging part is to generate the WHERE clause. Essentially, the issue with a sequence-to-sequence decoder is that the prediction of the next token depends on all previously generated tokens. However, different constraints may not have a dependency on each other. In our approach, SQLNet employs the sketch to provide the dependency relationship of different slots so that the prediction for each slot is only based on the predictions of other slots that it depends on. To implement this idea, the design of SQLNet introduces two novel constructions: *sequence-to-set* and *column attention*. The first is designed to predict an unordered set of constraints instead of an ordered sequence, and the second is designed to capture the dependency relationship defined in the sketch when predicting.

We evaluate our approach on the WikiSQL dataset (Zhong et al., 2017), which is, to the best of our knowledge, the only large scale NL2SQL dataset, and compare with the state-of-the-art approach, Seq2SQL (Zhong et al., 2017). Our approach results in the exact query-match accuracy of $61.5\%$ and the result-match accuracy of $68.3\%$ on the WikiSQL testset. In other words, SQLNet can achieve exact query-match and query-result-match accuracy of $7.5$ points and $8.9$ points higher than the corresponding metrics of Seq2SQL respectively, yielding the new state-of-the-art on the WikiSQL dataset.

The WikiSQL dataset was originally proposed to ensure that the training set and test set have a disjoint set of tables. In the practical setting, it is more likely that such an NL2SQL solution is deployed where there exists at least one query observed in the training set for the majority of tables. We re-organize the WikiSQL dataset to simulate this case and evaluate our approach and the baseline approach, Seq2SQL. We observe that in such a case the advantage of SQLNet over Seq2SQL enlarges by 2 points, and the SQLNet model can achieve an execution accuracy of $70.1\%$.

To summarize, our main contributions in this work are three-fold. First, we propose a novel principled approach to handle the sequence-to-set generation problem. Our approach avoids the "order-matters" problems in a sequence-to-sequence model and thus avoids the necessity to employ a reinforcement learning algorithm, and achieves a better performance than existing sequence-to-sequence-based approach. Second, we propose a novel attention structure called *column attention*, and show that this helps to further boost the performance over a raw sequence-to-set model. Last, we design SQLNet which bypasses the previous state-of-the-art approach by 9 to 13 points on the WikiSQL dataset, and yield the new state-of-the-art on an NL2SQL task.

## 2   SQL Query Synthesis from Natural Language Questions and Table Schema

In this work, we consider the WikiSQL task proposed in (Zhong et al., 2017). In particular, the input contains two parts: a natural language question stating the query for a table, and the schema of the table being queried. The schema of a table contains both the name and the type (i.e., real numbers or strings) of each column. The output is a SQL query which reflects the natural language question with respect to the queried table.

| **Table** | | | | | |
|---|---|---|---|---|---|
| **Player** | **No.** | **Nationality** | **Position** | **Years for Jazz** | **School/Club Team** |
| Stu Lantz | 22 | United States | Guard | 1974-75 | Nebraska |
| Rusty LaRue | 5 | United States | Guard | 2001-02 | Wake Forest |
| Eric Leckner | 45 | United States | Forward-Center | 1988-90 | Wyoming |
| Ron Lee | 18 | United States | Guard | 1979-80 | Oregon |
| Jim Les | 25 | United States | Guard | 1988-89 | Bradley |

**Question:**

Which country is Jim Les from?

**SQL:**

SELECT Nationality
WHERE Player = Jim Les

**Result:**

United States

Figure 1: An example of the WikiSQL task.

Note that the WikiSQL task considers synthesizing a SQL query with respect to only one table. Thus, in an output SQL query, only the SELECT clause and the WHERE clause need to be predicted, and the FROM clause can be omitted. We present an example in Figure 1.

The WikiSQL task makes further assumptions to make it tractable. First, it assumes that each column name is a meaningful natural language description so that the synthesis task is tractable from only the natural language question and column names. Second, any token in the output SQL query is either a SQL keyword or a sub-string of the natural language question. For example, when generating a constraint in the WHERE clause, e.g., name=`Bob`, the token `Bob` must appear in the natural language question as a sub-string. This assumption is necessary when the content in a database table is not given as an input. Third, each constraint in the WHERE clause has the form of COLUMN OP VALUE, where COLUMN is a column name, OP is one of "$<, =, >, \geq, \leq$", and VALUE is a substring of the natural language question as explained above.

Although these constraints seem to overly simplify the problem, we argue that such a subset of SQL queries still has a significant impact in practice. In fact, Johnson et al. (2017) studied 8.1 million real-world SQL queries written by Uber's data analysts, and found that at least 37% of them involve only one table and all constraints in the WHERE clause involve only one column. Thus, solving this problem can significantly reduce the burden of data analysts by reducing more than 1/3 of the workload in practice.

More importantly, different from most previous NL2SQL datasets (Tang & Mooney, 2001; Price, 1990; Dahl et al., 1994; Li & Jagadish, 2014; Pasupat & Liang, 2015; Yin et al., 2015), the WikiSQL task has several properties that we would like. First, it provides a large-scale dataset so that a neural network can be effectively trained. Note, previously famous data sets such as (Dahl et al., 1994) contain less than 10,000 examples; but modern deep learning models, e.g., a sequence-to-sequence model, typically require much more data. WikiSQL mitigates this issue by providing a larger dataset. Second, it employs crowd-sourcing to collect the natural language questions created by human beings, so that it can help to overcome the issue that a well-trained model may overfit to template-synthesized descriptions.

Third, we are interested in the SQL synthesis problem in an enterprise setting. In such a setting, the database may contain billions of users' sensitive information, and thus how to handle the scalability of the data and how to ensure privacy are important problems. Therefore, we prefer a solution that synthesizes SQL queries from only the natural language description and the table schema. Such requirements make many existing studies on question-answering (e.g., Sun et al. (2016)) and table content-based SQL synthesis approaches (e.g., Yin et al. (2015)) unsuitable. In this sense, the WikiSQL task perfectly fits our requirement to mitigate the scalability and privacy issues.

Fourth, the data is split so that the training, dev, and test set do not share tables. This helps to evaluate an approach's capability to generalize to an unseen schema. Previous datasets may have one or more of such four properties, but to the best of our knowledge, we are not aware of any NL2SQL dataset having all these properties except WikiSQL.

Besides these benefits, WikiSQL is still a challenging task. Zhong et al. (2017) report that the state-of-the-art task-agnostic semantic parsing model (Dong & Lapata, 2016) can achieve an execution accuracy of merely 37%, while the prevous state-of-the-art model for this task can achieve an execution accuracy of around 60%. Therefore, we believe tackling the WikiSQL task is a meaningful

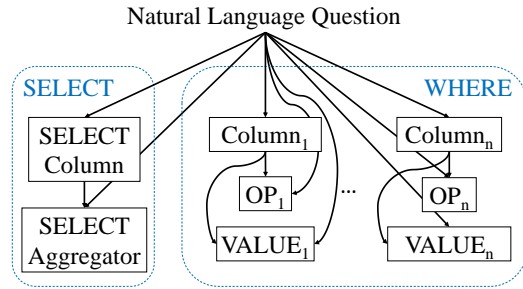

```
SELECT  $AGG $COLUMN
WHERE   $COLUMN $OP $VALUE
 (AND  $COLUMN $OP $VALUE) *
```

(a) SQL Sketch

(b) Graphical illustration of the dependency in a sketch

Figure 2: Sketch syntax and the dependency in a sketch

and challenging first step toward eventually solving the NL2SQL problem. We consider building and tackling the SQL synthesis task of more complex queries as important future work.

# 3 SQLNet

In this section, we present our SQLNet solution to tackle the WikiSQL task. Different from existing semantic parsing models (Dong & Lapata, 2016) which are designed to be agnostic to the output grammar, our basic idea is to employ a *sketch*, which highly aligns with the SQL grammar. Therefore, SQLNet only needs to fill in the *slots* in the sketch rather than to predict both the output grammar and the content.

The sketch is designed to be generic enough so that all SQL queries of interest can be expressed by the sketch. Therefore, using the sketch does not hinder our approach's generalizability. We will explain the details of a sketch in Section 3.1.

The sketch captures the dependency of the predictions to make. By doing so, the prediction of the value of one slot is only conditioned on the values of those slots that it depends on. This avoids the "order matters" problem in a sequence-to-sequence model, in which one prediction is conditioned on all previous predictions (Vinyals et al., 2016). To make predictions based on a sketch, we develop two techniques, sequence-to-set and column attention. We will explain the details of these techniques in Section 3.2.

We combine all techniques to design a SQLNet neural network to synthesize a SQL query from a natural language question and a table schema. In Section 3.3, we present the details of SQLNet and training details to surpass previous state-of-the-art approach without using reinforcement learning.

## 3.1 SKETCH-BASED QUERY SYNTHESIS

The SQL sketch that we employ is formally stated in Figure 2a. The tokens in bold (i.e., **SELECT**, **WHERE**, and **AND**) indicate the SQL keywords. The tokens starting with "$" indicate the *slot* to be filled. The name following the "$" indicates the type of the prediction. For example, the $AGG slot can be filled with either an empty token or one of the aggregation operators, such as SUM and MAX. The $COLUMN and the $VALUE slots need be filled with a column name and a sub-string of the question respectively. The $OP slot can take a value from $\{=, <, >\}$. The notion $(...)*$ employ a regular expression to indicate zero or more **AND** clauses.

The dependency graph of the sketch is illustrated in Figure 2b. All slots whose values are to be predicted are illustrated as boxes, and each dependency is depicted as a directed edge. For example, the box of $OP_1$ has two incoming edges from $Column_1$ and the natural language question respectively. These edges indicate that the prediction of the value for $OP_1$ depends on both the values of $Column_1$ and the natural language question. We can view our model as a graphical model based on this dependency graph, and the query synthesis problem as an inference problem on the graph. From this perspective, we can see that the prediction of one constraint is independent with another, and

thus our approach can fundamentally avoid the "order-matters" problem in a sequence-to-sequence model.

Note that although it is simple, this sketch is expressive enough to represent all queries in the WikiSQL task. Our SQLNet approach is not limited to this sketch only. To synthesize more complex SQL queries, we can simply employ a sketch that supports a richer syntax. In fact, the state-of-the-art approach on the WikiSQL task, i.e., Seq2SQL (Zhong et al., 2017), can also be viewed as a sketch-based approach. In particular, Seq2SQL predicts for $AGG and $COLUMN separately from the WHERE clause. However, Seq2SQL generates the WHERE clause using a sequence-to-sequence model. Thus it still suffers the "order-matters" problem.

## 3.2   SEQUENCE-TO-SET PREDICTION USING COLUMN ATTENTION

In this section, we use the prediction of a column name in the WHERE clause as an example to explain the ideas of a *sequence-to-set* model and *column attention*. We will explain the full SQLNet model in Section 3.3.

**Sequence-to-set.**   Intuitively, the column names appearing in the WHERE clause constitute a subset of the full set of all column names. Therefore, instead of generating a sequence of column names, we can simply predict which column names appear in this subset of interest. We refer to this idea as *sequence-to-set prediction*.

In particular, we compute the probability $P_{\mathbf{wherecol}}(col|Q)$, where $col$ is a column name and $Q$ is the natural language question. To this aim, one idea is to compute $P_{\mathbf{wherecol}}(col|Q)$ as

$$P_{\mathbf{wherecol}}(col|Q) = \sigma(u_c^T E_{col} + u_q^T E_Q) \tag{1}$$

where $\sigma$ is the sigmoid function, $E_{col}$ and $E_Q$ are the embeddings of the column name and the natural language question respectively, and $u_c$ and $u_q$ are two column vectors of trainable variables. Here, the embeddings $E_{col}$ and $E_Q$ can be computed as the hidden states of a bi-directional LSTM running on top of the sequences of $col$ and $Q$ respectively. Note the two LSTMs to encode the column names and the question do not share their weights. The dimensions of $u_c, u_q, E_{col}, E_Q$ are all $d$, which is the dimension of the hidden states of the LSTM.

In doing so, the decision of whether or not to include a particular column in the WHERE clause can be made independently to other columns by examining $P_{\mathbf{wherecol}}(col|Q)$.

**Column attention.**   Equation (1) has a problem of using $E_Q$. Since it is computed as the hidden states of the natural language question only, it may not be able to remember the particular information useful in predicting a particular column name. For example, in the question in Figure 1, the token "number" is more relevant to predicting the column "No." in the WHERE clause. However, the token "player" is more relevant to predicting the "player" column in the SELECT clause. The embedding should reflect the most relevant information in the natural language question when predicting on a particular column.

To incorporate this intuition, we design the *column attention* mechanism to compute $E_{Q|col}$ instead of $E_Q$. In particular, we assume $H_Q$ is a matrix of $d \times L$, where $L$ is the length of the natural language question. The $i$-th column of $H_Q$ represents the hidden states output of the LSTM corresponding to the $i$-th token of the question.

We compute the attention weights $w$ for each token in the question. In particular, $w$ is a $L$-dimension column vector, which is computed as

$$w = \mathbf{softmax}(v) \qquad v_i = (E_{col})^T W H_Q^i \quad \forall i \in \{1, ..., L\}$$

where $v_i$ indicates the $i$-th dimension of $v$, $H_Q^i$ indicates the $i$-th column of $H_Q$, and $W$ is a trainable matrix of size $d \times d$.

After the attention weights $w$ are computed, we can compute $E_{Q|col}$ as the weighted sum of each token's LSTM hidden output based on $w$:

$$E_{Q|col} = H_Q w$$

We can replace $E_Q$ with $E_{Q|col}$ in Equation (1) to get the column attention model:

$$P_{\mathbf{wherecol}}(col|Q) = \sigma(u_c^T E_{col} + u_q^T E_{Q|col}) \tag{2}$$

In fact, we find that adding one more layer of affine transformation before the $\sigma$ operator can improve the prediction performance by around $1.5\%$. Thus, we get the final model for predicting column names in the `WHERE` clause:

$$P_{\mathbf{wherecol}}(col|Q) = \sigma((u_a^{col})^T \mathbf{tanh}(U_c^{col} E_{col} + U_q^{col} E_{Q|col})) \tag{3}$$

where $U_c^{col}$ and $U_q^{col}$ are trainable matrices of size $d \times d$, and $u_a^{col}$ is a $d$-dimensional trainable vector.

We want to highlight that column attention is a special instance of the generic attention mechanism to compute the attention map on a question conditioned on the column names. We will show in our evaluation that this mechanism can improve upon a sequence-to-set model by around 3 points.

### 3.3   SQLNet MODEL AND TRAINING DETAILS

In this section, we present the full **SQLNet** model and training details. As illustrated in Figure 2b, the predictions of the `SELECT` clause and `WHERE` clause are separated. In the following, we first present the model for generating the `WHERE` clause and then the `SELECT` clause. In the end, we describe more training details which significantly help to improve the prediction accuracy.

#### 3.3.1   PREDICTING THE WHERE CLAUSE

The `WHERE` clause is the most complex structure to predict in the WikiSQL task. Our **SQLNet** model first predicts the set of columns that appear in the `WHERE` clause based on Section 3.2, and then for each column it generates the constraint by predicting the `OP` and `VALUE` slots. We describe them below.

**Column slots.**   After $P_{\mathbf{wherecol}}(col|Q)$ is computed based on Equation (3), **SQLNet** needs to decide which columns to include in the `WHERE`. One approach is to set a threshold $\tau \in (0, 1)$, so that all columns with $P_{\mathbf{wherecol}}(col|Q) \geq \tau$ are chosen.

However, we find that an alternative approach can typically give a better performance. We now explain this approach. In particular, we use a network to predict the total number $K$ of columns to be included in the subset, and choose the top-$K$ columns with the highest $P_{\mathbf{wherecol}}(col|Q)$ to form the column names in the `WHERE` clause.

We observe that most queries have a limited number of columns in their `WHERE` clauses. Therefore, we set an upper-bound $N$ on the number of columns to choose, and thus we cast the problem to predict the number of columns as a $(N+1)$-way classification problem (from 0 to $N$). In particular, we have

$$P_{\#\mathbf{col}}(K|Q) = \mathbf{softmax}(U_1^{\#\mathrm{col}} \mathbf{tanh}(U_2^{\#\mathrm{col}} E_{Q|Q}))_i$$

where $U_1^{\#\mathrm{col}}$ and $U_2^{\#\mathrm{col}}$ are trainable matrices of size $(N+1) \times d$ and $d \times d$ respectively. The notion $\mathbf{softmax}(...)_i$ indicates the $i$-th dimension of the softmax output, and we will use this notion throughout the rest of the description. **SQLNet** chooses the number of columns $K$ that maximizes $P_{\#\mathbf{col}}(K|Q)$.

In our evaluation, we simply choose $N = 4$ to simplify our evaluation setup. But note that we can get rid of the hyper-parameter $N$ by employing a variant-length prediction model, such as the one for the `SELECT` column prediction model that will be discussed in Section 3.3.2.

**`OP` slot.**   For each column in the `WHERE` clause, predicting the value of its `OP` slot is a 3-way classifications: the model needs to choose from $\{=, >, <\}$. Therefore, we compute

$$P_{\mathrm{op}}(i|Q, col) = \mathbf{softmax}(U_1^{\mathrm{op}} \mathbf{tanh}(U_c^{op} E_{col} + U_q^{op} E_{Q|col}))_i$$

where $col$ is the column under consideration, $U_1^{\mathrm{op}}, U_c^{op}, U_q^{op}$ are trainable matrices of size $3 \times d$, $d \times d$, and $d \times d$ respectively. Note that $E_{Q|col}$ is used in the right-hand side. This means that **SQLNet** uses column attention for `OP` prediction to capture the dependency in Figure 2b.

**VALUE slot.** For the `VALUE` slot, we need to predict a substring from the natural language question. To this end, SQLNet employs a sequence-to-sequence structure to generate the sub-string. Note that, here the order of the tokens in the `VALUE` slot indeed matters. Therefore, using a sequence-to-sequence structure is reasonable.

The encoder phase still employs a bi-directional LSTM. The decoder phase computes the distribution of the next token using a pointer network (Vinyals et al., 2015a; Yang et al., 2016) with the column attention mechanism. In particular, consider the hidden state of the previously generated sequence is $h$, and the LSTM output for each token in the natural language question is $H_Q^i$. Then the probability of the next token in `VALUE` can be computed as

$$P_{\text{val}}(i|Q, col, h) = \textbf{softmax}(a(h))$$

$$a(h)_i = (u^{\text{val}})^T \textbf{tanh}(U_1^{\text{val}} H_Q^i + U_2^{\text{val}} E_{col} + U_3^{\text{val}} h) \quad \forall i \in \{1, ..., L\}$$

where $u_a^{\text{val}}$ is a $d$-dimensional trainable vector, $U_h^{\text{val}}, U_c^{\text{val}}, U_q^{\text{val}}$ are three trainable matrices of size $d \times d$, and $L$ is the length of the natural language question. Note that the computation of the $a(h)_i$ is using the column attention mechanism, which is similar in the computation of $E_{Q|col}$.

Note that $P_{\text{val}}(i|Q, col, h)$ represents the probability that the next token to generate is the $i$-th token in the natural language question.SQLNet simply chooses the most probable one for each step to generate the sequence. Note that the ⟨END⟩ token also appears in the question. The SQLNet model stops generating for `VALUE` when the ⟨END⟩ token is predicted.

### 3.3.2 PREDICTING THE SELECT CLAUSE

The `SELECT` clause has an aggregator and a column name. The prediction of the column name in the `SELECT` clause is quite similar to the `WHERE` clause. The main difference is that for the `SELECT` clause, we only need to select one column among all. Therefore, we compute

$$P_{\textbf{selcol}}(i|Q) = \textbf{softmax}(sel)_i$$

$$sel_i = (u_a^{\text{sel}})^T \textbf{tanh}(U_c^{\text{sel}} E_{col_i} + U_q^{\text{sel}} E_{Q|col_i}) \quad \forall i \in \{1, ..., C\}$$

Here, $u_a^{\text{sel}}, U_c^{\text{sel}}, U_q^{\text{sel}}$ are similar to $u_a^{\text{col}}, U_c^{\text{col}}, U_q^{\text{col}}$ in (3), and $C$ is the total number of columns. Notice that each different dimension of the vector $sel$ is computed based on a corresponding column $col_i$. The model will predict the column $col_i$ that maximizes $P_{\textbf{selcol}}(i|Q)$.

For the aggregator, assuming the predicted column name for the `SELECT` clause is $col$, we can simply compute

$$P_{\textbf{agg}}(i|Q, col) = \textbf{softmax}(U^{\text{agg}} \textbf{tanh}(U_a E_{Q|col}))_i$$

where $U_a$ is a trainable matrix of size $6 \times d$. Notice that the prediction of the aggregator shares a similar structure as `OP`.

### 3.3.3 TRAINING DETAILS

In this section, we present more details to make our experiments reproducible. We also emphasize on the details that can improve our model's performance.

**Input encoding model details.** Both natural language descriptions and column names are treated as a sequence of tokens. We use the Stanford CoreNLP tokenizer (Manning et al., 2014) to parse the sentence. Each token is represented as a one-hot vector and fed into a word embedding vector before feeding them into the bi-directional LSTM. To this end, we use the GloVe word embedding (Pennington et al., 2014).

**Training details.** We need a special loss to train the sequence-to-set model. Intuitively, we design the loss to reward the correct prediction while penalizing the wrong prediction. In particular, given a question $Q$ and a set of $C$ columns $col$, assume $y$ is a $C$-dimensional vector where $y_j = 1$ indicates that the $j$-th column appears in the ground truth of the `WHERE` clause; and $y_j = 0$ otherwise. Then we minimize the following weighted negative log-likelihood loss to train the sub-model for $P_{\textbf{wherecol}}$:

$$loss(col, Q, y) = -\left( \sum_{j=1}^{C} (\alpha y_j \log P_{\textbf{wherecol}}(col_j|Q) + (1 - y_j) \log(1 - P_{\textbf{wherecol}}(col_j|Q)) \right)$$

In this function, the weight $\alpha$ is hyper-parameter to balance the positive data versus negative data. In our evaluation, we choose $\alpha = 3$. For all other sub-module besides $P_{\mathbf{wherecol}}$, we minimize the standard cross-entropy loss.

We choose the size of the hidden states to be 100. We use the Adam optimizer (Kingma & Ba, 2014) with a learning rate 0.001. We train the model for 200 epochs and the batch size is 64. We randomly re-shuffle the training data in each epoch.

**Weight sharing details.** The model contains multiple LSTMs for predicting different slots in the sketch. In our evaluation, we find that using different LSTM weights for predicting different slots yield better performance than making them share the weights. However, we find that sharing the same word embedding vector helps to improve the performance. Therefore, different components in SQLNet only share the word embedding.

**Training the word embedding.** In Seq2SQL, Zhong et al. (2017) suggest that the word embedding for tokens appearing in GloVe should be fixed during training. However, we observe that the performance can be boosted by 2 points when we allow the word embedding to be updated during training. Therefore, we initialize the word embedding with GloVe as discussed above, and allow them to be trained during the Adam updates after 100 epochs.

## 4 EVALUATION

In this section, we evaluate SQLNet versus the state-of-the-art approach, i.e., Seq2SQL (Zhong et al., 2017), on the WikiSQL dataset.

In the following, we first present the evaluation setup. Then we present the comparison between our approach and Seq2SQL on the query synthesis accuracy, as well as a break-down comparison on different sub-tasks. In the end, we propose another variant of the WikiSQL dataset to reflect another application scenario of the SQL query synthesis task and present our evaluation results of our approach versus Seq2SQL.

### 4.1 EVALUATION SETUP

In this work, we focus on the WikiSQL dataset (Zhong et al., 2017). The dataset was updated on October 16, 2017. In our evaluation, we use the updated version.

We compare our work with Seq2SQL, the state-of-the-art approach on the WikiSQL task. We compare SQLNet with Seq2SQL using three metrics to evaluate the query synthesis accuracy:

1. **Logical-form accuracy.** We directly compare the synthesized SQL query with the ground truth to check whether they match each other. This metric is used in (Zhong et al., 2017).

2. **Query-match accuracy.** We convert the synthesized SQL query and the ground truth into a canonical representation and compare whether two SQL queries match exactly. This metric can eliminate the false negatives due to only the ordering issue.

3. **Execution accuracy.** We execute both the synthesized query and the ground truth query and compare whether the results match to each other. This metric is used in (Zhong et al., 2017).

We are also interested in the break-down results on different sub-tasks: (1) the aggregator in the SELECT clause; (2) the column in the SELECT clause; and (3) the WHERE clause. Due to the different structure, it is hard to make a further fine-grained comparison.

We implement SQLNet using PyTorch (Facebook, 2017). For the baseline approach in our comparison, i.e., Seq2SQL, we compare our results with the numbers reported by Zhong et al. (2017).

However, Zhong et al. (2017) do not include the break-down results for different sub-tasks, and the source code is not available. To solve this issue, we re-implement Seq2SQL by ourselves. For evaluations whose results are not reported in (Zhong et al., 2017), we report the results from our re-implementation and compare SQLNet against those as the baseline.

|  | dev | | | test | | |
|---|---|---|---|---|---|---|
|  | $\text{Acc}_{\text{lf}}$ | $\text{Acc}_{\text{qm}}$ | $\text{Acc}_{\text{ex}}$ | $\text{Acc}_{\text{lf}}$ | $\text{Acc}_{\text{qm}}$ | $\text{Acc}_{\text{ex}}$ |
| Seq2SQL (Zhong et al. (2017)) | 49.5% | - | 60.8% | 48.3% | - | 59.4% |
| Seq2SQL (ours) | 52.5% | 53.5% | 62.1% | 50.8% | 51.6% | 60.4% |
| SQLNet | **-** | **63.2%** | **69.8%** | **-** | **61.3%** | **68.0%** |

Table 1: Overall result on the WikiSQL task. $\text{Acc}_{\text{lf}}$, $\text{Acc}_{\text{qm}}$, and $\text{Acc}_{\text{ex}}$ indicate the logical form, query-match and the execution accuracy respectively.

|  | dev | | | test | | |
|---|---|---|---|---|---|---|
|  | $\text{Acc}_{\text{agg}}$ | $\text{Acc}_{\text{sel}}$ | $\text{Acc}_{\text{where}}$ | $\text{Acc}_{\text{agg}}$ | $\text{Acc}_{\text{sel}}$ | $\text{Acc}_{\text{where}}$ |
| Seq2SQL (ours) | 90.0% | 89.6% | 62.1% | 90.1% | 88.9% | 60.2% |
| Seq2SQL (ours, C-order) | - | - | 63.3% | - | - | 61.2% |
| SQLNet (Seq2set) | - | - | 69.1% | - | - | 67.1% |
| SQLNet (Seq2set+CA) | **90.1%** | 91.1% | 72.1% | **90.3%** | 90.4% | 70.0% |
| SQLNet (Seq2set+CA+WE) | **90.1%** | **91.5%** | **74.1%** | **90.3%** | **90.9%** | **71.9%** |

Table 2: Break down result on the WikiSQL dataset. Seq2SQL (C-order) indicates that after Seq2SQL generates the WHERE clause, we convert both the prediction and the ground truth into a canonical order when being compared. Seq2set indicates that the sequence-to-set technique is employed. +CA indicates that column attention is used. +WE indicates that the word embedding is allowed to be trained. $\text{Acc}_{\text{agg}}$ and $\text{Acc}_{\text{sel}}$ indicate the accuracy on the aggregator and column prediction accuracy on the SELECT clause, and $\text{Acc}_{\text{where}}$ indicates the accuracy to generate the WHERE clause.

## 4.2 EVALUATION ON THE WIKISQL TASK

Table 1 presents the results for query synthesis accuracy of our approach and Seq2SQL. We first observe that our re-implementation of Seq2SQL yields better result than that reported in (Zhong et al., 2017). Since we do not have access to the source code of the original implementation, we cannot analyze the reason.

We observe that SQLNet outperforms Seq2SQL (even our version) by a large margin. On the logical-form metric, SQLNet outperforms our re-implementation of Seq2SQL by 10.7 points on the dev set and by 10.5 points on the test set. These advancements are even larger to reach 13.7 points and 13.0 points respectively if we compare with the original results reported in (Zhong et al., 2017). Note that even if we eliminate the false negatives of Seq2SQL by considering the query-match accuracy, the gap is only closed by 1 point, and still remains as large as 9.7 points. We attribute the reason to that Seq2SQL employs a sequence-to-sequence model and thus suffers the "order-matters" problem, while our sequence-to-set-based approach can entirely solve this issue.

On the execution accuracy metric, SQLNet is better than Seq2SQL (reported in Zhong et al. (2017)) by 9.0 points and 8.6 points respectively on the dev and test sets. Although they are still large, the advancements are not as large as those on the query-match metric. This phenomenon shows that, for some of the queries that Seq2SQL cannot predict exactly correct, (e.g., maybe due to the lack of one constraint in the WHERE clause), the execution results are still correct. We want to highlight that the execution accuracy is sensitive to the data in the table, which contributes to the difference between query-match accuracy and execution accuracy.

## 4.3 A BREAK-DOWN ANALYSIS ON THE WIKISQL TASK

We would like to further analyze SQLNet's and Seq2SQL's performance on different sub-tasks as well as the improvement provided by different techniques in SQLNet. The results are presented in Table 2.

We observe that on the SELECT clause prediction, the accuracy is around 90%. This shows that the SELECT clause is less challenging to predict than the WHERE clause. SQLNet's accuracy on the

| **Question:** | **SQL:** |
|---|---|
| **When twente came in third place and ajax was the winner what are the seasons?** | **SELECT season** 
 **WHERE winner = ajax AND third place = twente** |

Figure 3: An example of disordered pairs.

`SELECT` column prediction better than Seq2SQL. We attribute this improvement to the reason that **SQLNet** employs column attention.

We observe that the biggest advantage of **SQLNet** over Seq2SQL is on the `WHERE` clause's prediction accuracy. The improvement on the `WHERE` clause prediction is around 11 points to 12 points. Notice that the order of the constraints generated by Seq2SQL matters. To eliminate this effect, we evaluate the accuracy based on a canonical order, i.e., Seq2SQL (ours, C-order), in a similar way as the query-match accuracy. This metric will improve Seq2SQL's accuracy by 1 point, which obeys our observation on the overall query-match accuracy of Seq2SQL. However, we still observe that the **SQLNet** can outperform Seq2SQL by a large margin. From the break-down analysis, we can observe that the improvement from the usage of a sequence-to-set architecture is the largest to achieve around 6 points. The column attention further improves a sequence-to-set only model by 3 points, while allowing training word embedding gives another 2 points' improvement.

Note that the improvement on the `SELECT` prediction is around 2 points. The improvements from two clauses add up to the 13 points to 14 points improvements in total.

### 4.4 THE "ORDER-MATTERS" EFFECT

In this section, we document the "order-matters" phenomenon in the WikiSQL dataset, and study its impact on the Seq2SQL model. In WikiSQL, the constraints in the `WHERE` clause are listed in the ascending order of the column ID. We say a description-query pair is *disordered* if a clause in the description corresponding to a larger column ID appears earlier than another one corresponding to a smaller column ID. One example is shown in Figure 3. Note, in this example, a user is also likely to ask "When ajax was the winner and twente came in third place what are the seasons?". Thus, no matter how we assign column IDs, one of the two descriptions is disordered with respect to the SQL query.

We calculate the number of disordered pairs as follows. In fact, for each clause $col\ op\ val$, since $val$ must appear in the description as a substring, we can locate its first appearance in the description. By examining these locations for different constraints matches the order of the column IDs, we can examine whether a the description-query pair is disordered or not. From the WikiSQL dataset, we can calculate the percentage of disordered pairs as follows:

|  | training | dev | test |
|---|---|---|---|
| % of disordered samples | 6.2% | 6.5% | 6.2% |
| % of non-disordered samples ($\geq$ 2 columns) | 23.7% | 24.4% | 24.7% |
| % of samples with zero or one column | 70.1% | 69.1% | 69.1% |

In this table, we split all samples into three parts: (1) disordered samples; (2) non-disordered samples involving at least 2 columns in the `WHERE` clause; and (3) the rest containing at most one column in the `WHERE` clause. In fact, the second third does not cause the "order-matters" problem at all. Although we observe non-disordered samples are majority, the disordered ones constitute a non-negligible portion, i.e., $> 6\%$. Therefore, this portion may likely cause the "order-matters" problem.

We then examine whether the disordered samples indeed contributes to the "order-matters" phenomenon, and whether our proposed seq2set technique helps to mitigate the issue. To do this, we evaluate the effectiveness of Seq2SQL (our implementation) and the **SQLNet** (using sequence-to-set only) on the three parts. We train both models on the entire dataset. Then we split the dev set and the test set into three parts in the same way as above.

We report the results in Table 3. We observe that our sequence-to-set technique improves the performance on the disordered subset by a larger margin (i.e., $9.5\%$ to $16.5\%$) on both the dev set and

| | | disordered | | non-disordered ($\geq$ 2 col) | | rest | |
|---|---|---|---|---|---|---|---|
| | | $\text{Acc}_{\text{qm}}$ | $\text{Acc}_{\text{ex}}$ | $\text{Acc}_{\text{qm}}$ | $\text{Acc}_{\text{ex}}$ | $\text{Acc}_{\text{qm}}$ | $\text{Acc}_{\text{ex}}$ |
| dev | Seq2SQL (ours) | 19.5% | 33.6% | 35.9% | 53.9% | 62.8% | 67.5% |
| | SQLNet (Seq2set) | 33.1% | 43.1% | 44.0% | 55.4% | 65.3% | 69.7% |
| | Improvement | **13.6%** | **9.5%** | **8.1%** | **1.5%** | **2.5%** | **2.2%** |
| test | Seq2SQL (ours) | 18.8% | 34.5% | 33.4% | 50.1% | 61.1% | 66.5% |
| | SQLNet (Seq2set) | 35.3% | 46.2% | 43.1% | 54.0% | 63.7% | 68.9% |
| | Improvement | **16.5%** | **11.7%** | **9.7%** | **3.9%** | **2.6%** | **2.4%** |

Table 3: Evaluation results on disordered set versus non-disordered set. $\text{Acc}_{\text{qm}}$, and $\text{Acc}_{\text{ex}}$ indicate the query-match and the execution accuracy respectively.

| | dev | | | test | | |
|---|---|---|---|---|---|---|
| | $\text{Acc}_{\text{lf}}$ | $\text{Acc}_{\text{qm}}$ | $\text{Acc}_{\text{ex}}$ | $\text{Acc}_{\text{lf}}$ | $\text{Acc}_{\text{qm}}$ | $\text{Acc}_{\text{ex}}$ |
| Seq2SQL (ours) | 54.5% | 55.6% | 63.8% | 54.8% | 55.6% | 63.9% |
| SQLNet | - | **65.5%** | **71.5%** | - | **64.4%** | **70.3%** |

Table 4: Overall result on the WikiSQL variant dataset.

the test set than on the other two parts. This observation clearly shows that our sequence-to-set technique can mitigate the "order-matters" effect.

On the other hand, we observe that on the non-disordered set with two or more columns in the `WHERE` constraints, the $\text{Acc}_{\text{qm}}$ scores are also boosted by more than $8$ percentage points. We also attribute this to the "order-matters" effect: the disordered samples in the training set actually hurt the performance of a Seq2SQL model on the non-disordered samples; and since the sequence-to-set technique does not suffer from the "order-matters" issue at all, its advantage over Seq2SQL is large. The $\text{Acc}_{\text{ex}}$ is relatively small, i.e., $1.5\%$ to $3.9\%$. We observe that for some descriptions the Seq2SQL model actually generates some queries with fewer constraints in the `WHERE` clause than the ground truth; but since both queries produce the same results, they are counted as correct when computing $\text{Acc}_{\text{ex}}$.

Finally, on all samples with one or less constraints in the `WHERE` clause, the improvements of **SQL-Net** over Seq2SQL is not significant. Such samples do not have the "order-matters" issue at all. We attribute the improvement to the fact that **SQLNet** generates the constraint following the sketch, which leverages more grammar information than a sequence-to-sequence model used in Seq2SQL. This result provides further evidence to show that the improvement on the other two parts are mainly related to the "order-matters" issue.

## 4.5 EVALUATION ON A VARIANT OF THE WIKISQL TASK

In practice, a machine learning model is frequently retrained periodically to reflect the latest dataset. Therefore, it is more often that when a model is trained, the table in the test set is already seen in the training set. The original WikiSQL dataset is split so that the training, dev, and test sets are disjoint in their sets of tables, and thus it does not approximate this application scenario very well.

To better understand different model's performance in this alternative application scenario, we re-shuffle the data, so that all the tables appear at least once in the training set.

On this new dataset, we evaluate both **SQLNet** and Seq2SQL, and the results are presented in Table 4. We observe that all metrics of both approaches are improved. We attribute this to that all tables in the test set are observed by the models in the training set. This observation meets our expectation. The improvement of **SQLNet** over Seq2SQL (our implementation) remains the same across different metrics.

## 5 RELATED WORK

The study of translating natural language into SQL queries has a long history (Warren & Pereira, 1982; Androutsopoulos et al., 1993; 1995; Popescu et al., 2003; 2004; Li et al., 2006; Giordani & Moschitti, 2012; Zhang & Sun, 2013; Li & Jagadish, 2014; Wang et al., 2017). Earlier work focuses on specific databases and requires additional customization to generalize to each new database.

Recent work considers mitigating this issue by incorporating users' guidance (Li & Jagadish, 2014; Iyer et al., 2017). In contrast, SQLNet does not rely on human in the loop. Another direction incorporates the data in the table as an additional input (Pasupat & Liang, 2015; Mou et al., 2016). We argue that such an approach may suffer scalability and privacy issues when handling large scale user databases.

SQLizer (Yaghmazadeh et al., 2017) is a related work handling the same application scenario. SQL-izer is also a sketch-based approach so that it is not restricted to any specific database. Different from our work, SQLizer (Yaghmazadeh et al., 2017) relies on an off-the-shelf semantic parser (Berant et al., 2013; Manning et al., 2014) to translate a natural language question into a sketch, and then employs programming language techniques such as type-directed sketch completion and automatic repairing to iteratively refine the sketch into the final query. Since SQLizer does not require database-specific training and its code is not available, it is unclear how SQLizer will perform on the WikiSQL task. In this work, we focus on neural network approaches to handle the NL2SQL tasks.

Seq2SQL (Zhong et al., 2017) is the most relevant work and achieves the state-of-the-art on the WikiSQL task. We use Seq2SQL as the baseline in our work. Our SQLNet approach enjoys all the benefits of Seq2SQL, such as generalizability to an unseen schema and overcoming the inefficiency of a sequence-to-sequence model. Our approach improves over Seq2SQL in that by proposing a sequence-to-set-based approach, we eliminate the sequence-to-sequence structure when the order does not matter, so that we do not require reinforcement learning at all. These techniques enable SQLNet to outperform Seq2SQL by 9 points to 13 points.

The problem to parse a natural language to SQL queries can be considered as a special instance to the more generic semantic parsing problem. There have been many works considering parsing a natural language description into a logical form (Zelle & Mooney, 1996; Wong & Mooney, 2007; Zettlemoyer & Collins, 2007; 2012; Artzi & Zettlemoyer, 2011; 2013; Cai & Yates, 2013; Reddy et al., 2014; Liang et al., 2011; Quirk et al., 2015; Chen et al., 2016). Although they are not handling the SQL generation problem, we observe that most of them need to be fine-tuned to the specific domain of interest, and may not generalize.

Dong & Lapata (2016) provide a generic approach, i.e., a sequence-to-tree model, to handle the semantic parsing problem, which yields the state-of-the-art results on many tasks. This approach is evaluated in (Zhong et al., 2017), and it has been demonstrated less effective than the Seq2SQL approach. Thus, we do not include it in our comparison.

Some earlier approaches Wong & Mooney (2007) propose using Synchronous Context-Free Grammars (SCFG) to construct a semantic parser. However, a production rule in a SCFG is essentially a translation rule from the source language (e.g., English) to the target (e.g., SQL). Therefore, such an approach requires much more efforts to construct a SCFG and is less robust. On the other hand, our sketch is only a subset of SQL, which is simple to derive from the full SQL specification. Thus, our approach is more practical than previous SCFG-based approaches. Some recent works Rabinovich et al. (2017) propose incorporation grammar information during program generation. These approaches are similar to the Seq2SQL baseline that we have compared in our paper. In fact, we consider Seq2SQL as a special version of Rabinovich et al. (2017) that incorporates pointer networks for WHERE clause generation and reinforcement learning to boost the performance.

## 6 CONCLUSION

In this paper, we propose an approach, SQLNet, to handle an NL2SQL task. We observe that all existing approaches employing a sequence-to-sequence model suffer from the "order-matters" problem when the order does not matter. Previous attempts using reinforcement learning to solve this issue bring only a small improvement, e.g., by around 2 points. In our work, SQLNet fundamentally solves the "order-matters" problem by employing a sequence-to-set model to generate SQL queries

when order does not matter. We further introduce the column attention mechanism, which can further boost a sequence-to-set model's performance. In total, we observe that our SQLNet system can improve over the prior art, i.e., Seq2SQL, by a large margin ranging from 9 points to 13 points on various metrics. This demonstrates that our approach can effectively solve the "order-matters" problem, and shed new light on novel solutions to structural generation problems when order does not matter.

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
