# OpenReview forum: "SQLNet: Generating Structured Queries From Natural Language Without Reinforcement Learning"
_ICLR.cc/2018/Conference — Reject_

### Official Review · AnonReviewer3 · 2017-11-27
**The motivation of the work is neither convincing nor verified empirically**

**Rating:** 4
**Confidence:** 5

**Review:**

This submission proposes a new seq2sel solution by adopting two new techniques, a sequence-to-set model and column attention mechanism. They show performance improve over existing studies on WikiSQL dataset.

While the paper is written clearly, the contributions of the work heavily depends on the WikiSQL dataset. It is not sure if the approach is generally applicable to other sequence-to-sql workloads. Detailed comments are listed below:

1. WikiSQL dataset contains only a small class of SQL queries, with aggregation over single table and various filtering conditions. It does not involve any complex operator in relational database system, e.g., join and groupby. Due to its simple structure, the problem of sequence-to-sql translation over WikiSQL is actually simplified as a parameter selection problem for a fixed template. This greatly limits the generalization of approaches only applicable to WikiSQL. The authors are encouraged to explore other datasets available in the literature.

2. The "order-matters" motivation is not very convincing. It is straightforward to employ a global ordering approach to rank the columns and filtering conditions based on certain rules, e.g., alphabetical order. That could ensure the orders in the SQL results are always consistent.

3. The experiments do not fully verify how the approaches bring performance improvements. In the current version, the authors only report superficial accuracy results on final outcomes, without any deep investigation into why and how their approach works. For instance, they could verify how much accuracy improvement is due to the insensitivity to order in filtering expressions.

4. They do not compare against state-of-the-art solution on column and expression selection. While their attention mechanism over the columns could bring performance improvement, they should have included experiments over existing solutions designed for similar purpose. In (Yin, et al., IJCAI 2016), for example, representations over the columns are learned to generate better column selection.

As a conclusion, I find the submission contains certain interesting ideas but lacks serious research investigations. The quality of the paper could be much enhanced, if the authors deepen their studies on this direction.

---

> ### Author Response · Authors · 2017-12-25
> **Response to AnonReviewer3**
>
> We thank the reviewers for the valuable comments. We would like to clarify some clear misunderstandings and highlight the differences in our revisions.
>
> We agree with the reviewer that this work focus on the WikiSQL dataset. This is because this is the only largest scale dataset that is close to a practical application scenario to the best of our knowledge. In our revision, we cite one recent case study on 8.1 million real-world SQL queries written by uber data analysts (Johnson 2017). They show that almost 40% of all these queries (1) involve only table; and (2) each WHERE constraint involves only one column. They do not contain join at all. This is exactly the same as the queries proposed in WikiSQL. On the other hand, this problem is not trivial, since we can see even our new state-of-the-art’s performance is less than 70%. By showing these two points, we believe we are dealing with a meaningful problem which is not trivial to tackle.
>
> To all other datasets, such as atis-3 mentioned by Reviewer A or the dataset used in (Yin, et al., IJCAI 2016), they suffer one or more problems discussed in Section 2 which render them not practical, and thus not an ideal target of our study.
>
> We include one more section in our evaluation to document the order-matters phenomenon. In particular, we want to emphasize that WikiSQL is already employing a global ordering, but the columns may appear in the natural language statement in an arbitrary order. For example, we include the following example in our revision:
> NL:  What are the seasons when twente came in third place and ajax was the winner?
> SQL: SELECT season WHERE winner = ajax AND third place = twente
>
> We can observe that the statement “twente came in third place” and “ajax was the winner” appear in the reverse order of the global order of the two columns “winner” and “third place”. This is the ``order-matters” issue we discuss. However, this issue cannot be solved by changing the global order of the two columns, since the human users should also be allowed to state:
> “What are the seasons when ajax was the winner and twente came in third place?”
> No matter what global order is used, one of these two statements will cause the ``order-matters” issue. As far as we can see, the only way to mitigate this issue is to restrict human users to state their goals following the global order. Again, doing so will render the dataset artificial and not practical.
>
> We are very confused about the reviewer’s comment “The experiments do not fully verify how the approaches bring performance improvements”. In fact, the entire Section 4.3 is devoted to an ablation study to show the improvements brought by each component. In particular, the SQLNet (seq2set) shows the accuracy improvement due to the insensitivity to order. In our revision, we add Sec 4.4 to provide one more section to even further understand the effectiveness due to “order-matters” issue.
>
> Seq2SQL is the state-of-the-art on the WikiSQL dataset, and we have compared against it. Yin et al 2016 is not suitable for the WikiSQL task since their approach needs to take the data in the table as a part of the input. We have argued that this is not a scalable approach and may also have privacy issue. We have discussed this in Section 2.
>
> We hope the reviewer can clarify some of the earlier comments with respect to our clarification. We are also welcome more comments.

---

### Official Review · AnonReviewer2 · 2017-11-28
**Well motivated and straightforward approach for WikiSQL**

**Rating:** 7
**Confidence:** 4

**Review:**

The authors present a neural architecture for the WikiSQL task. The approach can be largely seen as graphical model tailored towards the constrained definition of SQL queries in WikiSQL. The model makes strong independence-assumptions, and only includes interactions between structures where necessary, which reduces the model complexity while alleviating the "order matters" problem. An attention mechanism over the columns is used to model the interaction between columns and the op or value in a soft differentiable manner. The results show impressive gains over the baseline, despite using a much simpler model. I appreciated the breakdown of accuracy over the various subtasks, which provides insights into where the challenges lie.

---

> ### Author Response · Authors · 2017-12-25
> **Response to AnonReviewer2**
>
> We thank the reviewer’s comment. We have updated the paper to address some comments raised in all reviews. We have posted a separate comment for a highlight overview of revision, and updated the paper. Please take a look and see if there are any comments that we should address further. More feedbacks are welcome!

---

### Official Review · AnonReviewer1 · 2017-11-28
**Re-positioning with respect to the literature is needed**

**Rating:** 5
**Confidence:** 5

**Review:**

This paper proposes a neural network-based approach to converting natural language questions to SQL queries. The idea is to use a small grammar to facilitate the process, together making some independence assumptions. It is evaluated on a recently introduced dataset for natural language to SQL.

Pros:
- good problem, NL2SQL is an important task given how dominant SQL is
- incorporating a grammar ("sketch") is a sensible improvement.

Cons:
- The dataset used makes very strong simplification assumptions. Not  problem per se, but it is not the most challenging SQL dataset. The ATIS corpus is NL2SQL and much more challenging and realistic:
Deborah A. Dahl, Madeleine Bates, Michael Brown, William Fisher, Kate Hunicke-Smith, David Pallett, Christine Pao, Alexander Rudnicky, and Elizabeth Shriberg. 1994. Expanding the scope of the ATIS task: the ATIS-3 corpus. In Proceedings of the workshop on Human Language Technology (HLT '94). Association for Computational Linguistics, Stroudsburg, PA, USA, 43-48. DOI: https://doi.org/10.3115/1075812.1075823

- In particular, the assumption that every token in the SQL statement is either an SQL keyword or appears in the natural language statement is rather atypical and unrealistic.

- The use of a grammar in the context of semantic parsing is not novel; see this tutorial for many pointers:
http://yoavartzi.com/tutorial/

- As far as I can tell, the set prediction is essentially predicted each element independently, without taking into account any dependencies. Nothing wrong, but also nothing novel, that is what most semantic parsing/semantic role labeling baseline approaches do. The lack of ordering among the edges, doesn't mean they are independent.

- Given the rather constrained type of questions and SQL statements, it would make sense to compare it against approaches for question answering over knowledge-bases:
https://github.com/scottyih/Slides/blob/master/QA%20Tutorial.pdf
While SQL can express much more complex queries, the ones supported by the grammar here are not very different.

- Pasupat and Liang (2015) also split the data to make sure different tables appear only in training, dev, test and they developed their dataset using crowd sourcing.

- The comparison against Dong and Lapata (2016) is not fair because their model is agnostic and thus applicable to 4 datasets while the one presented here is tailored to the dataset due the grammar/sketch used. Also, suggesting that previous methods might not generalize well sounds odd given that the method proposed seems to use much larger datasets.

- Not sure I agree that mixing the same tables across training/dev/test is more realistic. If anything, it assumes more training data and manual annotation every time a new table is added.

---

> ### Author Response · Authors · 2017-12-25
> **Response to AnonReviewer1**
>
> We appreciate reviewers’ valuable comments, and we have improved our paper to address some of the concerns. We find that most comments on the novelty are to some points that we do not claim as our contribution (e.g., the WikiSQL task itself is not our contribution at all). We clarify some of such confusions below, and hope the reviewers can provide more feedback to help us to improve our paper.
>
> First, the reviewer mentioned ATIS-3. We agree with the reviewer that atis-3 is much more challenging than WikiSQL. But we want to emphasize that we choose the problem not simply based on its difficulty, but also based on its practical impact. In our revision, we cite one recent case study on 8.1 million real-world SQL queries written by uber data analysts (Johnson 2017). They show that almost 40% of all these queries (1) involve only one table; and (2) each WHERE constraint involves only one column. This is exactly the same as the queries proposed in WikiSQL. On the other hand, this problem is not trivial, since we can see even our new state-of-the-art’s performance is less than 70%. By showing these two points, although we are not solving a challenging problem as 'NP vs P', we believe we are dealing with a meaningful problem which is not trivial to tackle.
>
> On the other hand, although atis-3 is more challenging, we observe that its dataset is small by the deep neural network standard. This is one additional reason why we prefer WikiSQL.
>
> The reviewer mentioned: “In particular, the assumption that every token in the SQL statement is either an SQL keyword or appears in the natural language statement is rather atypical and unrealistic.” We want to emphasize that this is NOT true. We only assume the value in the query must appear in the description to make the problem amenable, but we do not assume the column names appear in the description.
>
> For the constraints on the values, we agree that further efforts need to devote to making a better dataset. But we do not see the problem is overly simplified as discussed above.
>
> Next, the reviewer mentioned: “The use of a grammar in the context of semantic parsing is not novel”. We agree with the reviewers, and we also didn’t claim using a grammar is our contribution. At the end of the introduction, we highlight the three contributions of this work: (1) seq2set; (2) column attention; (3) achieving the state-of-the-art on WikiSQL.
>
> We do not follow very clearly about the comments “the set prediction is essentially predicted each element independently, without taking into account any dependencies”. Clearly, predicting the value in one constraint in the WHERE clause depends on the column selected in a previous step. Also, it is mentioned that “nothing novel” and “this is most … baselines do”. We would highly appreciate it if the reviewer could provide some references. To the best of our knowledge, some typical baseline approaches such as Seq2tree has been demonstrated ineffective on this Wikisql dataset in Zhong et al. 2017.
>
> The reviewer mentioned QA tasks. First, since QA has been studied over decades and many QA tasks have been proposed (and mentioned in the slides), we would appreciate it if the reviewer can point out the particular one that is relevant. Second, in our understanding, most existing works on KB-based QA will take the entire KB as an input to answer a question. We have argued in our paper why we do not prefer such a problem: the KB can be too huge or contain privacy-sensitive information, and thus generating the query without touching the data itself is an important factor for practical usages. Again, WikiSQL task is more suitable to such a requirement than previously proposed tasks involving data itself.
>
> For the overnight dataset (Pasupat and Liang 2015), it is not true that the schemas from train/dev/test are non-overlapping. We quote the statement from (Pasupat and Liang 2015):
> “For each domain, we held out a random 20% of the examples as the test set, and performed development on the remaining 80%, further splitting it to a training and development set (80%/20%). We created a database for each domain by randomly generating facts using entities and properties in the domain (with type-checking).”
> Here, each domain is one schema. Also, the novelty of WikiSQL over Overnight is not the problem that we want to address in our paper.
>
> We are not comparing against Seq2tree (Dong et al 2016), which was originally compared in Zhong et al 2017. We only compare to the Seq2SQL which is the state-of-the-art on the WikiSQL dataset. Again, as we discussed above, our work is focusing on the WikiSQL dataset itself, since we believe that is an important, though somehow narrow, task.
>
> Johnson et al,  Practical differential privacy for SQL queries using elastic sensitivity. to appear in VLDB 2017.

---

> > ### Comment · AnonReviewer1 · 2017-12-28
> > **Response to author response (part 1)**
> >
> > I appreciate the thoughtful response by the authors. Here are some comments/feedback:
> >
> > - WikiSQL/ATIS-3: I appreciate that WikiSQL is not your contribution. However it is your choice to work on this task and not others, and thus the choice and conclusions drawn from the experiments need to be argued with precision. In the version of the paper submitted for reviewing, it read that WikiSQL was considered more challenging than others previously considered. Now this statement has been revised appropriately. Furthermore, the fact that the methods developed in this paper require larger training datasets than those applied to ATIS is rather a disdavantage in the context of semantic parsing research. Finally it should be stated in the paper explicitly that the WikiSQL is not only a larger dataset but also a narrower task, as stated in the response.
> >
> > - "The reviewer mentioned: “In particular, the assumption that every token in the SQL statement is either an SQL keyword or appears in the natural language statement is rather atypical and unrealistic.” We want to emphasize that this is NOT true."
> >
> > In both the original and the revised version it reads: "Second, any token in the output SQL query
> > is either a SQL keyword or a sub-string of the natural language question." I assumed that column names are included in the tokens of the SQL statement. In the example of figure 1, it seems like the only token from the SQL statement that doesn't appear in the NL question is "no.", but that's only if we assume the system doesn't know that "number" mean "no.", which fairly trivial to learn given the large-scale training dataset. GeoQuery has the same kind of challenge and it was less than 1000 instances, but nevertheless performances reached 80% accuracy for Enlgish using just phrase-based machine translation, see: https://people.eecs.berkeley.edu/~jda/papers/avc_smt_semparse.pdf. Could you provide a more challenging example from WikiSQL to help clarify the challenge posed by it?
> >
> > - "We agree with the reviewers, and we also didn’t claim using a grammar is our contribution.": Indeed, but neither in the original paper nor in the revised you give credit to any previous work. You don't claim you invented seq2seq either, but you give credit to those who proposed it (as you should). Given that you mention "sketch" 39 times in your paper, grammar based semantic parsing seems to be also important enough previous work to be acknowledged and cited.
> >
> > -  "Clearly, predicting the value in one constraint in the WHERE clause depends on the column selected in a previous step. Also, it is mentioned that “nothing novel” and “this is most … baselines do”. We would highly appreciate it if the reviewer could provide some references. To the best of our knowledge, some typical baseline approaches such as Seq2tree has been demonstrated ineffective on this Wikisql dataset in Zhong et al. 2017."
> >
> > Here is a paper from 2006 that employs a grammar (a CFG in particular) that given one rule generating two intermediate nodes, each of them is explanded independently:
> > http://www.mt-archive.info/HLT-NAACL-2006-Wong.pdf
> > If what you do is different, it would be good to compare against this paper, as well as others beyond Zhong et al. (2017).
> >
> > - "First, since QA has been studied over decades and many QA tasks have been proposed (and mentioned in the slides), we would appreciate it if the reviewer can point out the particular one that is relevant. "
> >
> > Here is one that seems to be handling the same kind of questions:
> > http://cs.ucsb.edu/~ysu/papers/www16_table.pdf
> > They cite a fair amount of previous work that is also worth considering.
> >
> > -  "We have argued in our paper why we do not prefer such a problem: the KB can be too huge or contain privacy-sensitive information, and thus generating the query without touching the data itself is an important factor for practical usages."
> >
> > The paper I cite above handles KBs with millions of tables, thus it definitely scales to WikiSQL size databases. I would argue that being able to handle large databases is desirable, and hence an advantage for the methods that can do it. Secondly, creating the query without "touching the data itself" is only possible when the NL question contains the values, an assumption made in the WikiSQL dataset but not necessarily realistic. When it doesn't hold, some form of (named) entity linking is necessary. I appreciate the privacy concerns and it would be worth stating precisely which aspects of the previous work such as the one mentioned above violates them, since they have different variants of their system utilizing different aspects of the data.

---

> > > ### Comment · AnonReviewer1 · 2017-12-28
> > > **Response to author response (part 2)**
> > >
> > > - "For the overnight dataset (Pasupat and Liang 2015), it is not true that the schemas from train/dev/test are non-overlapping. We quote the statement from (Pasupat and Liang 2015):"
> > >
> > > I think there is a misunderstanding. The overnight dataset was poposed in this work:
> > > https://nlp.stanford.edu/pubs/wang-berant-liang-acl2015.pdf
> > > (Wang, Berant and Liang, ACL 2015)
> > > The Pasupat and Liang (ACL 2015 too) is this one:
> > > https://cs.stanford.edu/~ppasupat/resource/ACL2015-paper.pdf
> > > In this one, we read:
> > > "The final dataset contains 22,033 examples on 2,108 tables. We set aside 20% of the tables and
> > > their associated questions as the test set and develop on the remaining examples"
> > > Thus the tables (and their schemas) in the test set do not appear in the training/dev set.

---

> > > > ### Author Response · Authors · 2018-01-05
> > > > **Responses to feedbacks part2 from Reviewer A**
> > > >
> > > > We apologize for misunderstanding the comments. We are aware of the WikiTableQuestion task as well. However, it is a question-answering task, rather than a query-generation task. We have explained in previous responses why we prefer query-generation over touching the data directly. This is also why we thought the overnight dataset was referred to before, since overnight is a query-generation dataset.

---

> > > ### Author Response · Authors · 2017-12-29
> > > **Responses to feedbacks from Reviewer A (Part 1)**
> > >
> > > Thanks a lot for the further feedback! We have uploaded a further revision to reflect our responses. More feedbacks are welcome, and we would be happy to address as many of them as possible before the end of the discussion period.
> > >
> > > First, we have changed the example to the following one:
> > > Question: Which country is Jim Les from?
> > > SQL query: SELECT Nationality WHERE Player ＝ Jim Les
> > >
> > > In this example, the column name “nationality” is not a simple rename of any utterance in the description. There have been several others, for example:
> > > Question: What is the green house made of?
> > > SQL query: SELECT Composition WHERE Colours = green
> > >
> > > These examples should justify our argument that the column names are not a substring of the original statement. Note, our model can correctly process these examples.
> > >
> > > Second, “grammar based semantic parsing seems to be also important enough previous work to be acknowledged and cited.” We cite a few more papers:
> > >
> > > Maxim Rabinovich, Mitchell Stern, Dan Klein, Abstract Syntax Networks for Code Generation and Semantic Parsing, EMNLP 2017
> > > Parisotto, Emilio, et al. Neuro-symbolic program synthesis, ICLR 2017
> > >
> > > We think these are the most relevant approaches, which are quite similar to Seq2SQL baseline. In fact, we can view Seq2SQL as a modification of these works by incorporating the pointer network for value prediction. We are happy to cite more papers that the reviewer would suggest.
> > >
> > > Third, thanks for the pointer to the 2006 paper! However, we would like to argue that the SCFG used in (Wong 2006) is quite different than the sketch (or CFG) used in our work, and constructing an SCFG requires much more efforts.
> > >
> > > In fact, an SCFG production rule is of the form NT -> (A, B), where A and B indicate rules in the source and target language respectively. In the semantic parsing setting, A is the natural language and B is SQL. This means that such a rule is essentially a translation rule to say that every natural sentence A should be translated into B. Therefore, constructing an SCFG is essentially not only constructing the grammar B, but also constructing a formal grammar for natural language A and the translation rule from A to B, which is highly non-trivial.
> > >
> > > In our sketch-based approach, we only need the grammar B, but NEITHER the grammar for A NOR the translation rule from A to B. Further, grammar B is typically already available, since B is a programming language, i.e., SQL. Therefore, to apply our approach, we only need to construct a dependency graph for a SQL sub-grammar, which requires not many efforts. This makes our approach much more practical than SCFG-based approaches such as Wong (2006). We want to note that constructing an SCFG, which requires constructing a set of translation rules and a manually designed alignment approach (as in Wong 2006), is a non-trivial work, and we find it hard to justify why such an approach serves a more reasonable baseline than Zhong et al 2017.
> > >
> > > Having said this, we acknowledge the concerns from Reviewer A that more baselines should be compared. We will examine Seq2seq, seq2tree, and abstract syntax network, which are likely the state-of-the-arts on parsing and semantic parsing. The time may not be sufficient before the rebuttal period, but we guarantee that our best results of these approaches will be provided in the final version. Based on our existing experience, these approaches are unlikely to outperform Zhong et al 2017.

---

> > > > ### Author Response · Authors · 2017-12-29
> > > > **Responses to feedbacks from Reviewer A (Part 2)**
> > > >
> > > > Fourth, we acknowledge of the QA task studied in Sun et al 2016, and they are very important. However, as we mentioned, their approaches need to touch the data to propose candidate answers first and then rank the answers directly. However, the applications that we care most are in the enterprise setting dealing with user’s private data. One such scenario is described in the Uber study (Johnson et al. 2018). Therefore, we focus on query generation rather than question answering.
> > > >
> > > > Having said this, we acknowledge the concerns from the reviewer that such an approach will have the limitation that the values must appear as a substring of the description, and we have stated it clearly in Section 2. However, we argue that touching the data directly is not ideal due to the privacy concern. In our agenda, we plan to study the propose-repair scheme to mitigate this issue. That is, we can generate a query based on the description, and query the database, and revise the query. Such an approach is similar to the PL-approach such as SQLizer. Note that in this process, querying the database may still leak private information. However, we can apply the approach proposed by Johnson et al. 2018 to automatically convert a SQL query into a differentially private one so that the information leakage can be mitigated. Such a paradigm is much better than QA directly with respect to the privacy concern.
> > > >
> > > > With respect to the scalability issue, our target application scenarios may contain billions of records in a database (e.g., the Uber study in Johnson et al. 2018). A model handling such large-scale datasets typically special designs. In a SQL query synthesis setting, we do not think such a burden is necessary, and thus we prefer synthesizing SQL queries directly from description and schema, without touching the data. We want to also emphasize that WikiSQL does not reflect the scale of the target applications in our mind, but its problem setup automatically makes the database scale not an issue since, during query generation, the data in the table is not touched.

---

### Author Response · Authors · 2017-12-25
**Our latest revision**

We have improved our paper with the following revision:
We have added more discussions in Section 2 to explain why WikiSQL is a more meaningful and challenging task that is more practical than previous datasets, as part of our explanation why our work, dealing with WikiSQL, is a meaningful contribution.
We have added a separate subsection (Section 4.4) to document our study on the 'order-matters' issue, and we also explain why this is not a specific issue in WikiSQL but a general issue that may be encountered in all other tasks. We also provide more detailed analysis to show how our seq2set technique helps to mitigate this issue.

---

> ### Comment · AnonReviewer1 · 2018-01-12
> **Response to latest revision**
>
> I have looked at the latest revision and it has not addressed my main concern adequately, i.e. the positioning of the paper wrt the literature. It is argued that previous work on question answering over tables is not relevant to the paper, which is clearly not the case: even though SQL can express more complex queries than these, the dataset considered does exactly that by utilizing only a part of the expressivity of SQL. Given this, it doesn't acknowledge the fact that crowd sourced datasets already exist, which do not share tables in training/dev/test. In addition, it is argued that using a sketch (the well established notion of a grammar in previous work), doesn't hinder the generality of the approach while it clearly does, especially when comparing it against work agnostic to the task such as the one of Dong and Lapata (2016). Thus I keep my score unchanged.

---

### Decision · Program_Chairs · 2018-01-29
**ICLR 2018 Conference Acceptance Decision**

**Decision:**

Reject

**Comment:**

The pros and cons of the paper cited by the reviewers can be summarized as follows:

Pros:
- good problem, NL2SQL is an important task given how dominant SQL is
- incorporating a grammar ("sketch") is a sensible improvement.

Cons:
- The dataset used makes very strong simplification assumptions (that every token is an SQL keyword or appears in the NL)
- The use of a grammar in the context of semantic parsing is not novel, and no empirical comparison is made against other reasonable recent baselines that do so (e.g. Rabinovich et al. 2017).

Overall, the paper seems to do some engineering for the task of generating SQL, but without an empirical comparison to other general-purpose architectures that incorporate grammars in a similar way, the results seem incomplete, and thus I cannot recommend that the paper be accepted at this time.